Tamilokus mabinia, a new, anatomically divergent genus and species of wood-boring bivalve from the Philippines

Shipway J. Reuben 1
Altamia Marvin A. 1
Rosenberg Gary 2
Concepcion Gisela P. 3
Haygood Margo G. 4
Distel Daniel L. dan/distel@northeastern.edu 1
1 Ocean Genome Legacy Center, Department of Marine and Environmental Science, Northeastern University , Nahant , MA , United States of America
2 Academy of Natural Sciences, Drexel University , Philadelphia , PA , United States of America
3 Marine Science Institute, University of the Philippines , Quezon City , Philippines
4 Department of Medicinal Chemistry, University of Utah , Salt Lake City , UT , United States of America
Williams Suzanne
Electronic publication date: 2019 Feb 7
Publication date: 2019
Volume: 7
Electronic Location ID: e6256
Received 2018 Aug 15; Accepted 2018 Dec 10
Copyright: ©2019 Shipway et al.
Copyright year: 2019
Copyright holder: Shipway et al.
License: This is an open access article distributed under the terms of the Creative Commons Attribution License, which permits unrestricted use, distribution, reproduction and adaptation in any medium and for any purpose provided that it is properly attributed. For attribution, the original author(s), title, publication source (PeerJ) and either DOI or URL of the article must be cited.
License URL: https://creativecommons.org/licenses/by/4.0/

Keywords: Shipworm, Philippines, Phylogeny, New species, Teredinidae, Biodiversity, Micro-CT, New genus, Wood-Borer, Zoology

Funding: Fogarty International Center of the National Institutes of Health Award U19TW008163 National Science Foundation 1442759 1722553 The research reported herein was supported by Fogarty International Center of the National Institutes of Health Award U19TW008163 (to Margo G. Haygood, Gisela P. Concepcion, Gary Rosenberg and Daniel L. Distel), and by National Science Foundation Awards 1442759 and 1722553 (to Daniel L. Distel). The funders had no role in study design, data collection and analysis, decision to publish, or preparation of the manuscript.

==============================
Here we describe an anatomically divergent wood-boring bivalve belonging to the family Teredinidae. Specimens were collected off the coast of Mabini, Batangas, Philippines, in February 2018, from sunken driftwood at a depth of less than 2 m. A combination of characteristics differentiates these specimens from members of previously named teredinid genera and species. Most notable among these include: an enlarged cephalic hood which extends across the posterior slope of the shell valves and integrates into the posterior adductor muscle; a unique structure, which we term the ‘cephalic collar’, formed by protruding folds of the mantle immediately ventral to the foot and extending past the posterior margin of the valves; a large globular stomach located entirely posterior to the posterior adductor muscle and extending substantially beyond the posterior gape of the valves; an elongate crystalline style and style sac extending from the base of the foot, past the posterior adductor muscle, to the posteriorly located stomach; calcareous pallets distinct from those of described genera; a prominently flared mantle collar which extends midway along the stalk of the pallets; and, separated siphons that bear a pigmented pinstripe pattern with highly elaborate compound papillae on the incurrent siphon aperture. We used Micro-Computed Tomography (Micro-CT) to build a virtual 3D anatomical model of this organism, confirming the spatial arrangement of the structures described above. Phylogenetic analysis of the small (18S) and large (28S) nuclear rRNA gene sequences, place this bivalve within the Teredindae on a branch well differentiated from previously named genera and species. We propose the new genus and species Tamilokus mabinia to accommodate these organisms, raising the total number of genera in this economically and environmentally important family to 17. This study demonstrates the efficacy of Micro-CT for anatomical description of a systematically challenging group of bivalves whose highly derived body plans are differentiated predominantly by soft tissue adaptations rather than features of calcareous hard-parts.

Introduction

The Teredinidae, commonly referred to as shipworms, are a group of wood-boring bivalves with wide-ranging economic and ecological impacts in coastal marine systems. Their ability to bore into and digest wood is estimated to cause billions of dollars in damage per year to coastal structures, such as piers, jetties, wharfs, fishing equipment and aquaculture pens (Turner 1966; Distel 2003). Yet this same ability allows them to play a fundamental role in carbon cycling by processing recalcitrant organic carbon trapped in wood and liberating nutrients and energy that would otherwise be less accessible to other organisms (Nishimoto et al., 2015; Charles et al., 2016).

The unique wood-boring life-style of the Teredinidae has led to several anatomical specialisations and as such, this family is believed to exhibit more variation in morphology than any other group in the Mollusca (Turner, 1988). In contrast to more typical bivalves, the shell is greatly reduced in size and has prominent pedal and posterior gapes, allowing the protrusion of the foot and elongate body respectively. The location of the shell valves at the extreme anterior end of the body covers little of the visceral mass and offers little or no protection to the animal. Instead, the shell, which features tiny denticulated ridges on its surface, functions as an effective drilling and grinding tool for burrowing into wood (Nair, 1966; Turner, 1966; Voight, 2015). The body is long, vermiform and enclosed by a calcareous tube that is secreted onto the burrow walls, surrounding the animal from the posterior edge of the valves to the burrow opening. The siphons are flanked by a pair of calcareous, paddle or feather-like structures, known as the pallets. These pallets, which are unique to Teredinidae, serve as watertight and predator-resistant plugs to seal the burrow entrance (Nair, 1966; Turner, 1966; Voight, 2015). Currently, there are 16 accepted genera composed of 70 species in the Teredinidae (Bulatov, 1933; Turner, 1966; Macintosh, 2012; Borges & Merckelbach, 2018; Velásquez & Shipway, 2018). However, it has been recently suggested that between 80-100 species likely exist (Huber, 2015; Shipway et al., 2016).

Using integrative taxonomic methods, including anatomical, morphological and molecular data (Borges et al., 2012; Shipway et al., 2016), we identify and describe a new genus and species of wood-boring teredinid from the Philippines, which we name Tamilokus mabinia.

Materials & Methods

Specimen collection

Nine specimens of Tamilokus mabinia were collected in February 2018 off the coast of Balayan Bay, Mabini, Batangas, Philippines (13.758°N, 120.925°E), at a depth of <2 m, from a log measuring approximately 2 m in total length, as part of the Philippine Mollusk Symbiont (PMS) International Collaborative Biodiversity Group (ICBG) expedition. The specimen location map (Fig. 1) was produced and approximate coordinates were determined using the National Oceanographic and Atmospheric Administration Map Viewer (https://www.nauticalcharts.noaa.gov/ENCOnline/enconline.html). Specimens were carefully extracted from their burrows, photographed (Fig. 2), measured, fixed in 4% formaldehyde solution freshly prepared from paraformaldehyde (PFA), washed, and dehydrated through an ethanol series (30%, 50% and 70%, 30 min per wash, x2 washes) with final storage in 70% ethanol. Partial specimens, identifiable as T. mabinia based on the unique morphology of the pallets, siphons and cephalic collar, were dissected or were fixed in 100% ethanol for DNA preservation. Morphological features (Fig. 3) and calcareous structures (Fig. 4) were imaged using the Keyence VHX-6000 Digital Microscope (Osaka, Japan). Specimens were directly compared with all described teredinid genera from the Harvard Museum of Comparative Zoology. Part of this work was completed under the supervision of the Department of Agriculture-Bureau of Fisheries and Aquatic Resources, Philippines (DA-BFAR) in compliance with all required legal instruments and regulatory issuances covering the conduct of the research. All Philippine specimens used in this study were obtained using Gratuitous Permit GP-0140-17 issued by DA-BFAR.

Figure 1 Specimen collection site.

(A) overview of the Philippines; (B) overview of Mabini, Batangas; (C) collection site. Scale bar = 0.5 Km.

Figure 2 Tamilokus mabinia.

(A) Holotype (PMS-3916Y); (B) paratype (PMS-3915X); (C) paratype (PMS-4051L). Ca, caecum; CC, cephalic collar; CH, cephalic hood; DG, digestive glands; Ft, foot; Gi, gill; In, Intestine; MC, mantle collar; Pa, pallet; Si, siphon; St, stomach; SV, shell valve. Scale bar = 1.0 cm.

Figure 3 Morphological characters of Tamilokus mabinia.

(A) Tamilokus mabinia major morphological characters (PMS-3949Y). Scale bar = 5.0 mm; (B–C) cephalic hood; (D–E) cephalic collar; (F) siphon features; (G) magnified region of the incurrent siphon. Scale bar =500 µm. CC, cephalic collar; CH, cephalic hood; ES, excurrent siphon; Ft, foot; IP, inner papillae; IS, incurrent siphon; MC, mantle collar; Pa, pallet; OP, outer papillae; Si, siphon.

Figure 4 Calcareous structures of Tamilokus mabinia.

(A) Inner surface and (B) outer surface of pallet. Specimen PMS-3943P. Scale bar = 1.0 mm; (C) shell valve outer surface. Specimen PMS-3915X. Scale bar = 500 µm. The valves are very thin and brittle. Note that the ventral tip of the valve is damaged. Anterior to left, dorsal is toward top.

Micro-computed tomography

Specimen PMS-4051L was stained for 20 days in 10% iodine prior to imaging. Imaging was performed using a SkyScan 1173 micro-CT scanner (Bruker Micro-CT, Kontich, Belgium) equipped with a Hamamatsu 130/300 tungsten X-ray source and a FlatPanel Sensor camera detector with 2,240 × 2,240 pixels. Scanning parameters were as follows: source voltage = 60 kV; source current = 100 µA; exposure time = 900 ms; frames averaged = 3; frames acquired over 180° = 960; filter = no; binning = no; flat field correction = activated; scanning time =(X3)00:28:14; number of connected scans= 3; and, random movement = ON (10). Reconstruction of the raw data was accomplished using the software provided with the scanner (NRecon 1.6.6.0, Bruker Micro-CT, Kontich, Belgium). The following settings were employed to enhance image contrast and to compensate for ring and streak artefacts; smoothing = no, ring artefact correction = 4, and beam hardening correction = activated. Reconstructed scans were then analyzed using CTVox (Bruker Micro-CT, Kontich, Belgium). A Micro-CT 3D rendered model of Tamilokus mabinia, including major anatomical structures and transverse cross sections of the model is shown in Fig. 5.

Figure 5 Microcomputed Tomography 3D render of Tamilokus mabinia.

(A) major anatomical structures; (B–M) transverse sections of A from anterior to posterior. AC, anal canal; Ca, caecum; CC, cephalic collar; CH, cephalic hood; CS, crystalline style; CSS, crystalline style sac; DG, digestive gland; EBV, efferent branchial vein; Ft, foot; Gi, gill; Go, gonad; He, heart; In, intestine; MC, mantle collar; MO, mouth; Oe, oesophagus; PAM, posterior adductor muscle; SCA; stomach-caecum aperture; Si, siphon; St, stomach; SV, shell valve. Specimen PMS-4051L. Scale bar = 2.5 mm.

DNA extraction & amplification

DNA was extracted from the siphonal tissue and associated musculature from one specimen of Tamilokus mabinia (PMS-3943P), using the DNeasy Blood & Tissue kit (Qiagen, Hilden, Germany) following manufacturer’s protocol. Approximate purity, concentration and yield of DNA were determined by UV spectrophotometry. Genomic DNA was cryo-preserved at −80 °C and archived at the Ocean Genome Legacy Center of New England Biolabs, Northeastern University, Nahant, MA, USA (accession number in Table 1). The small (18S) and large (28S) subunit nuclear rRNA genes were amplified from the resultant DNA preparation by polymerase chain reaction (PCR). Amplification reactions were prepared using 12.5 µL of high-fidelity polymerase solution (OneTaq, New England Biolabs, Ipswich, Massachusetts), 0.5 µL of each primer (10 mM), 1–2 µL DNA template (10–20 ng/µL), brought to a total volume of 25 µL with purified water. Fragments of the small (18S) and large (28S) subunit nuclear rRNA genes were amplified using the primer pairs 18S EukF (5′–WAY-CTG-GTT-GAT-CCT-GCC-AGT–3′) and 18S EukR (5′–TGA-TCC-TTC-YGC-AGG-TTC-ACC-TAC–3′) (Medlin et al., 1988); 28S-NLF184-21 (5′–ACC-CGC-TGA-AYT-TAA-GCA-TAT–3′) and 28S-1600R (5′–AGC-GCC-ATC-CAT-TTT-CAG-G–3′) (Distel et al., 2011), resulting in amplicons of approximately 1,686 and 1,416 base pairs, respectively. The PCR amplification proceeded as follows: an initial denaturation step of 94 °C for three minutes, followed by 35 cycles with a denaturation step of 94 °C for 20 s, an annealing step of 64 °C for 40 s for the 18S and 63 °C for 30 s for 28S, an extension step of 68 °C for 60 s and a final extension of 68 °C for five minutes. All reactions were performed on a PTC-200 Thermal Cycler (MJ Research, Quebec, Canada).

Table 1 Specimen details and collection site information.

Specimen number	Holding institution	Accession number	Type	Preparation	Figure	
PMS-3899P	MSI	PMS-3899P	Paratype	Preserved in 70% ethanol	–	
PMS-3915X	ANSP	A476699	Paratype	Preserved in 70% ethanol	Figs. 2B & 4B	
PMS-3916Ya	ANSP	A476699	Holotype	Preserved in 70% ethanol	Fig. 2A	
PMS-3943Pb	MSI	PMS-3943P	Paratype	Preserved in 70% ethanol	Figs. 4A & Fig. 6	
PMS-3943Pb	OGL	E28692	–	Genomic DNA voucher	Fig. 6	
PMS-3949Y	MSI	PMS-3949Y	Paratype	Preserved in 70% ethanol	Fig. 3	
PMS-4037Y	ANSP	A476699	Paratype	Fixed in 4% PFA, transferred to 70% ethanol	–	
PMS-4051L	ANSP	A476699	Paratype	Fixed in 4% PFA, transferred to 70% ethanol, stained with 10% iodine for Micro-CT	Figs. 2C & Fig. 5	
Notes.

a The holotype will be transferred to the National Museum of the Philippines pending completion of an ongoing reorganization.

b Partial specimen with missing valves.

For each template, three separate amplicons, all produced under identical conditions, were pooled, cleaned and concentrated using the Zymo Clean & Concentrator Kit (Irvine, CA). Resulting products were sequenced bidirectionally on a 3730xl DNA Analyzer (Life Technologies, Grand Island, NY) using the Big Dye Terminator 3.1 Cycle Sequencing Kit (Life Technologies, Grand Island, NY) at New England Biolabs (Ipswich, Massachusetts). Primary sequence data from the small (18S) and large (28S) subunit nuclear rRNA genes were submitted to GenBank (NCBI) under accession numbers MH974682 and MH974683, respectively.

Phylogenetic analysis

The 18S and 28S rRNA gene sequences from specimen PMS-3943P were concatenated and aligned as in Distel et al. (2011) with sequences representing 10 of 16 recognized genera (Bulatov, 1933; Turner, 1966) including: Bankia Gray 1842, Dicyathifer Iredale 1932, Kuphus Guettard 1770, Lyrodus Gould 1870, Nausitora Wright 1864, Neoteredo Bartsch 1920, Spathoteredo Moll 1928, Teredo Linnaeus 1758, Teredora Bartsch 1921, and Teredothyra Bartsch 1921. Phylogenetic analysis was performed using MrBayes version 3.2.6 (Ronquist & Huelsenbeck, 2003) implemented in Geneious version 10.2.2 (https://www.geneious.com) using GTR + I + Γ nucleotide substitution model. Placopecten magellanicus was specified as the outgroup, the chain length was set to 5 million, subsampling every 2,000 generations and discarding the first 20% of the results as burn-in.

Zoobank registration

The electronic version of this article in Portable Document Format (PDF) will represent a published work according to the International Commission on Zoological Nomenclature (ICZN), and hence the new names contained in the electronic version are effectively published under that Code from the electronic edition alone. This published work and the nomenclatural acts it contains have been registered in ZooBank, the online registration system for the ICZN. The ZooBank LSIDs (Life Science Identifiers) can be resolved and the associated information viewed through any standard web browser by appending the LSID to the prefix http://zoobank.org/. The LSID for this publication is: LSID urn:lsid:zoobank.org:pub:0235E48D-D7B7-4A4B-A078-4204C1907240. The online version of this work is archived and available from the following digital repositories: PeerJ, PubMed Central and CLOCKSS.

Results

Systematics

Family Teredinidae Rafinesque, 1815	
Tamilokus Shipway, Distel & Rosenberg, gen. nov.	
urn:lsid:zoobank.org:act:61B916B1-54D1-418D-B1A3-342162C261C2	

Type species: Tamilokus mabinia sp. nov.

Type material: Holotype PMS-3916Y; paratypes PMS-3899P, PMS-3915X, PMS-3943P, PMS-3949Y, PMS-4037Y, and PMS-4051L. Smallest to largest specimens measured 6.2 cm–15.4 cm in total body length. The holotype is currently held at the Academy of Natural Sciences of Philadelphia (ANSP) and will be deposited in the National Museum of the Philippines in Manila pending completion of an ongoing reorganization. Paratypes are deposited at ANSP, at the Marine Science Institute of the University of the Philippines (MSI) and at the Ocean Genome Legacy Center of New England Biolabs, Northeastern University (OGL). For catalog numbers assigned by holding institutions see Table 1.

Type locality: Balayan Bay, off Mabini, Batangas Province, Philippines (coordinates 13.758°N, 120.925°E).

Comparative Material: The following material was examined: Teredora malleolus (Turton, 1822) MCZ-350541 (3 specimens); Teredora princesae (Sivickis, 1928) MCZ-232090 (5 specimens); Uperotus clava (Gmelin, 1791) MCZ-238009 (3 specimens); and, Uperotus panamensis (Bartsch, 1922) MCZ-357824 (4 specimens).

Diagnosis: pallets triangular-shaped with an ovate, flattened stalk, prominent cephalic hood and cephalic crest, crystalline style extends beyond posterior adductor muscle, caecum U-shaped doubling back upon itself, siphons with distinct pin-striped pigmentation with incurrent siphon featuring three rows of papillae

Etymology: Tamilokus (masculine), in recognition of the common name for shipworm in the Philippines, ‘tamilok’.

Habitat: Marine, wood-borer.

Description: Pallets very small in relation to body length (Fig. 2A), formed of a broad, flattened, translucent and ovate stalk, with a solid, white calcareous blade (Fig. 4A); cephalic hood prominent, extending to cover the posterior slope of the shell valves (Figs. 2A, 3A–3C & 5A, 5C); “cephalic collar”, formed from protruding folds of the mantle, extending along the ventral surface from the base of the foot to a point slightly posterior to the posterior margin of the shell valves (Figs. 2A–2B, 3A, 3D–3E, 5A–5B); large crystalline style and style sac originating within the cephalic collar and extending beyond the posterior adductor muscle into the posteriorly located stomach (Figs. 5A–5E); large globular stomach (Figs. 2B & 5A, 5E–5F) located posteriorly in relation to posterior adductor muscle and valve pedal gape, composed of three lobes, with muscular opening into caecum (Figs. 5A, 5F); large digestive glands (Figs. 2A–2C & Fig. 5A, 5C–5E); caecum (Figs. 2A–2C & 5A, 5D–5I) large, elongate, U-shaped, doubling back upon itself anteriorly towards the right (Figs. 2B & 5D–5I), lacking a typhlosole (Figs. 5D–5I); intestine unusually broad and voluminous (Fig. 2B), especially where it passes ventral to the caecum (Figs. 5G–5I), containing short ovoid faecal pellets packed in multiple rows across its width (Fig. 2B); intestine extends anteriorly, looping over crystalline style sac (Figs. 5C–5E), then doubles back posteriorly, looping under the caecum before extending anteriorly on the dorsal surface of the caecum, finally looping under the posterior adductor muscle before extending a short distance posteriorly and opening into the anal canal; anal canal is open and does not retain faeces; gonad located centrally (Figs. 5A, 5I–5K), beginning posterior to the caecum and ending at the heart; heart located medially (Figs. 5J–5K); gill extends from the siphons anteriorly to the posterior tip of the gonads (Figs. 5A, 5K–5L); prominently flared mantle collar around siphons and pallets (Figs. 2B, 3A & 5A).

Remarks: Tamilokus gen. nov may be easily differentiated from all other genera within the family Teredinidae based on pallet morphology. The simple triangular cup-shaped pallet and thick ovate stalk of Tamilokus may be distinguished from the pallets of: Bankia Gray 1842, Nausitora Wright 1864, Nototeredo Bartsch 1923, and Spathoteredo Moll 1928 by the lack of segmentation; from Lyrodus Gould 1870, Teredo Linnaeus 1758, Zachsia Bulatoff & Rjabtschikoff 1933, and Nivanteredo Velásquez & Shipway, 2018 by the absence of a periostracum and presence of a broad ovate stalk; from Teredothyra Bartsch 1921 by the presence of a single undivided cup; from Dicyathifer Iredale 1932 and Kuphus Guettard 1770 by the absence of a medial ridge; from Bactronophorus Tapparone Canefri 1877 by the absence of a dagger-like extension; from Neoteredo Bartsch 1920, Psiloteredo Bartsch 1922, Teredora Bartsch 1921 and Uperotus Guettard 1770 by the cup-rather than paddle-shaped blades, and from the latter three genera by the lack a distinct thumb-nail like depression bearing concentric or radiating ridges.

Additionally, Tamilokus may be easily distinguished from Bactronophorus Tapparone Canefri 1877, Dicyathifer Iredale 1932, Neoteredo Bartsch 1920 and Teredothyra Bartsch 1921 by the absence of a muscular sphincter at the posterior end of the anal canal; from Lyrodus Gould 1870, Teredo Linnaeus 1758 and Zachsia Bulatoff & Rjabtschikoff 1933 by the absence of brood pouches on the gill; from Neoteredo Bartsch 1920 by the absence of dorsal lappets; and from Kuphus Guettard 1770 by the presence of a caecum and the absence of a strong muscular collar surrounding the valves.

Tamilokus is similar to Teredora Bartsch 1921 and Uperotus Guettard 1770 in that it possesses a U-shaped caecum that, after passing posteriorly from the stomach, doubles back upon itself towards the right and extends anteriorly, terminating near its origin on the right side of the stomach (Turner, 1966). However, Tamilokus can be easily distinguished from these genera based on pallet morphology (as previously described). Additionally, the gills of Teredora and Uperotus extend the entire length of the animal from the base of the siphons to the mouth, whereas those of Tamilokus terminate near the posterior end of the caecum; the labial palps in both Teredora and Uperotus are large and free, but are very small and attached in Tamilokus; the crystalline style in both Teredora and Uperotus is located anteriorly to the posterior adductor muscle, but extends from the base of the foot well beyond the posterior margin of the adductor muscle in Tamilokus; the stomach of both Teredora and Uperotus is located anterior to posterior adductor muscle, whereas the stomach of Tamilokus is located posterior to the posterior adductor muscle; the heart of both Teredora and Uperotus is positioned anteriorly, but is medially positioned in Tamilokus; the siphons of Teredora and Uperotus are united along their entire length, but are separate in Tamilokus; the incurrent siphons of both Teredora and Uperotus feature a single primary row of papillae, whereas the incurrent siphon of Tamilokus has an additional secondary row that includes compound branched papillae; and, the cephalic hood in both Teredora and Uperotus is inconspicuous, whereas the cephalic hood in Tamilokus is prominent and covers the posterior slope of the shell valves. Finally, Tamilokus may be recognized by the presence of the cephalic collar which is unique to this genus. Taxonomic characters differentiating Tamilokus from Teredora and Uperotus are summarized in Table 2.

Table 2 Diagnostic characters of all teredinid genera featuring a U-shaped ceacum.

Taxonomic character	Tamilokus	Teredora	Uperotus	
Labial palps	Reduced/absent	Free	Large and free	
Crystalline style	Extends from base of foot beyond posterior adductor muscle	Located anterior to posterior adductor muscle	Located anterior to posterior adductor muscle	
Stomach	Globular, located posterior to posterior adductor muscle	Globular, located anterior to posterior adductor muscle	Globular, located anterior to posterior adductor muscle	
Caecum	Doubles back upon itself, to the right	Doubles back upon itself, to the right	Doubles back upon itself, to the right	
Caecal typhlosole	Absent	Rudimentary	Absent	
Heart	Median position	Located anteriorly	Located anteriorly	
Gills	Located posteriorly, extending to posterior caecum	Extend from base of siphons to mouth	Extend from base of siphons to mouth	
Siphons	Separated along entire length, pink pinstriped pigmentation	United to the tip	United to the tip	
Incurrent siphon papillae	Primary row numerous small papillae, secondary row compound papillae	Incurrent siphon numerous large papillae	Incurrent siphon numerous small papillae	
Excurrent siphon papillae	Numerous small papillae	–	Excurrent siphon two large papillae on dorsal surface	
Cephalic hood	Prominent, covers valve posterior slope	Inconspicuous, does not cover valve posterior slope	Inconspicuous, does not cover valve posterior slope	
Cephalic collar	Present	Absent	Absent	
Flared mantle collar	Present	Absent	Absent	
Pallets	Triangular, cup-shaped, non-segmented, ovate flattened stalk	Paddle shaped, non-segmented, ‘thumbnail’ depression	Paddle shaped, non-segmented, ‘thumbnail’ depression, radiating ribs	

Tamilokus mabiniaShipway & Distel,sp. nov.	
urn:lsid:zoobank.org:act:88D193B6-9062-48C2-AC9D-EC6F858B3C05	

Etymology: Mabinia (noun in apposition), in honour of Apolinario Mabini, a Philippine national hero, and pertaining to the type location of the specimens from Mabini, Batangas, Philippines.

Description: All characteristics of the genus, plus; dorsal anterior slope of shell valves highly elongated (Fig. 4B); incurrent and excurrent siphons ringed with papillae (Fig. 3F); incurrent siphon features outer ring of long papillae, and inner ring of shorter compound or branched papillae (Fig. 3G); siphon tips are ringed with pink to brownish red pigment that extends posteriorly in narrow closely spaced parallel stripes (Figs. 2A–2C & 3A).

Definition: Pallets elongate, composed of an unsegmented blade built upon a central stalk (Fig. 4A); blade triangular, single concave U-shaped cup on distal margin with slightly more pronounced curvature on outer face; periostracum absent; mantle collar flared, extending approximately half the length of the pallet; stalk and blade approximately equal in length; stalk translucent, ovate and flattened in both sagittal and transverse section; blade calcareous and white.

Phylogeny

Bayesian analysis of concatenated small (18S) and large (28S) nuclear rRNA gene sequences (Fig. S1) indicate a basal position for Tamilokus within Teredinidae, sharing a well-supported node with Teredora, but separated from that taxon by a branch length comparable to those separating most accepted teredinid genera. A subtree excerpted from Fig. S1 is displayed in Fig. 6.

Figure 6 Phylogenetic position of Tamilokus mabinia among the Teredinidae.

(A) sub tree excerpted from a Bayesian analysis of the concatenated 18S and 28S nuclear rRNA gene sequences. The tree was constructed using the taxa presented in Distel et al. (2011). The full tree is presented in Fig. S1. Numbers at nodes indicate posterior probabilities.

Discussion

Herein, we describe a divergent member of the Teredinidae that, based on anatomical, morphological and molecular characteristics, is distinct from members of described genera. Among the distinguishing features of Tamilokus mabinia, the most striking are the posterior position of the stomach and the elongation of the crystalline style and style sac to accommodate this posterior location (Fig. 5A). This position of the stomach is found only in Tamilokus but is similar to that found in Kuphus, except that in the latter, the entire visceral mass is located posterior to the posterior adductor. The distinct positions of these two taxa on the phylogenetic tree of Teredinidae suggests that this similarity is due to convergence rather than homology. The extension of the crystalline style, from a position ventral to and near the anterior end of the foot to a position well posterior to the shell valves, is unique to Tamilokus (Turner, 1966). In other bivalves, cilia-mediated rotation of the crystalline style is thought to aid digestion by mixing enzymes and reducing food particle size through mechanical action (Robin et al., 2018; Nelson, 1918; Edmondson, 1920; Yonge, 1923; Lavine, 1946; Morton, 1952; Horiuchi & Lane, 1982; Alyakrinskaya, 2001; Sakamoto et al., 2008; Mackenzie & Marshall, 2014). However, the functional significance of the exceptionally large style of Tamilokus remains to be explained.

Tamilokus also features a characteristic structure absent in other genera, which we term the ‘cephalic collar’. This feature is formed by protruding folds of the mantle and is located directly ventral to the foot and anterior adductor muscle and extends posterior to the shell valves on the ventral mantle surface (Figs. 2A–2B, 3A, 3D-3E & 5A). Figures 3D–3E show that the cephalic collar can be contracted or expanded, and Micro-CT scans reveal that the style sac is located within the cephalic collar (Figs. 5A–5B). These observations suggest that the cephalic collar may serve to accommodate the large size of the crystalline style and may facilitate its movement.

Another unusual feature of Tamilokus is the caecum, which folds sharply to the right at its midpoint, doubling back upon itself to form a compressed U-shape. Among the Teredinidae, only two described genera, Teredora and Uperotus share this feature (Turner, 1966). However, Teredora and Uperotus differ from Tamilokus in several important ways. In these genera, the gills extend almost the entire length of the animal from the base of the siphons to the mouth, proportionately the longest gills among the family (Turner, 1966). In addition to the extended gills, the labial palps, which help sort and direct filtered particles to the mouth, are prominent and detached. This observation led to suggestions that these genera are specialized for planktotrophy rather than wood feeding (Turner, 1966; Nair & Saraswathy, 1971). This hypothesis, however, preceded the discovery of cellulolytic symbionts in the gills of shipworms, providing the competing hypothesis that larger gills are adaptive for wood digestion (Distel et al., 2011).

In contrast to Teredora and Uperotus, the gills of Tamilokus extend only to the posterior end of the caecum (Fig. 5A) and the palps are small, fixed, and appear to be vestigial. Also, in contrast to these genera, the intestine of Tamilokus is extremely wide. In all specimens examined, the capacious intestine was tightly packed with light-coloured excavated wood fiber (Fig. 2B), matching the colour of the wood substrate in which the animals burrowed. No evidence of planktonic organisms could be discerned in caecum contents or faecal material by microscopic examination.

Siphon morphology also distinguishes Tamilokus from Teredora and Uperotus. The siphons of the former are separate while those of the latter two are united along their entire length. The presence of pink pinstripes on both the incurrent and excurrent siphons of Tamilokus (Figs. 2 and 3) are also unique among described species. Additionally, the siphons are tipped with two rows of papillae on the incurrent siphon aperture; the primary outer row is simple in structure, while the inner secondary row include shorter compound papillae with multiple branches (Figs. 3F–3G). Characteristics of the siphons have been suggested as potentially useful in teredinid identification (Turner, 1966; Nair & Saraswathy, 1971), but have been described for only a small number of species (Roch, 1940; Turner, 1966; Lopes & Narchi, 1998; De Moraes & Lopes, 2003). Although the degree of genetic, ontogenetic and ecophenotypic variation in teredinid siphons has yet to be documented (Turner, 1966), characterizing these features could prove valuable for field identification, as specimens could be studied in situ without the necessity of extracting the teredinid from its burrow.

Aside from its unique anatomical and morphological features, the classification of Tamilokus as a new genus is further supported by molecular phylogenetic data. Phylogenetic analysis based on concatenated small (18S) and large (28S) nuclear rRNA gene sequences reveal well supported relationships between Tamilokus and basal taxa within Teredinidae. However, the branch length of Tamilokus is among the longest in the family, indicating divergence comparable to or greater than those separating accepted teredinid genera (Fig. 6). Anatomical and morphological characters are also consistent with the basal position of Tamilokus. As in other basal members of the family, the stomach of T. mabinia is globular (Type 2 of Turner 1966 (Turner, 1966)), the intestine loops anteriorly over the crystalline style sac, and brooded larvae are absent in the gills of all specimens examined. This is consistent with the hypothesis that these characteristics are plesiomorphic and that the elongate stomach, absence of the anterior intestinal loop and complex reproductive strategies are apomorphic (Distel et al., 2011).

In this investigation, Micro-CT has been invaluable for elucidating soft tissue anatomy. This has been especially important in this systematically challenging group, which is characterised by highly derived body parts and few taxonomically informative calcareous features. Previously, Micro-CT has been used to determine quantitative measurements of wood degradation and boring rates of Teredinidae (Amon et al., 2015). Additionally, Micro-CT has been used to reveal exceptionally well-preserved fossilized wood-boring bivalves in mid-Cretaceous wood, with silicified soft parts and overall body-plan characteristic of the Teredinidae (Robin et al., 2018). Herein, we further demonstrate the power of this technique to reveal the relationships among soft tissue features in three dimensions, and in the evaluation of likely mode of nutrition and life history strategy, while minimizing distortion that can result from manual dissection.

The extent of diversity within the Teredinidae has long been a subject of debate. Historically, as many as 130 species have been recognized (Moll & Roch, 1931), but many were later synonymized in the first comprehensive work on the taxonomy of this family (Turner, 1966). Recently, it has been suggested that teredinid diversity may be greater than previously thought (Huber, 2015; Shipway et al., 2016). This view is supported by the recent description of one new genus and several new species of teredinid bivalves (Macintosh, 2012; Borges & Merckelbach, 2018; Velásquez & Shipway, 2018). In addition, further reports suggest several more species (Shipway et al., 2016; Treneman et al., 2018) and at least one more genus (Lozouet & Plaziat, 2008) remain to be described.

Conclusions

We describe a new and anatomically divergent genus and species of wood-boring bivalve (Teredinidae) from the Philippines, which we name Tamilokus mabinia. A detailed Micro-CT 3D render of T. mabinia reveals several anatomically divergent characters, including a new structure which we term the ‘cephalic collar’, a prominent crystalline style sac and crystalline style originating within the cephalic collar and extending beyond the posterior adductor muscle into the posteriorly located stomach, and a caecum (wood-storing organ) which doubles back upon itself. In addition, the unique pallets, the primary taxonomic character of this family, could not be placed in any existing genera. Phylogenetic analysis places T. mabinia in a separate taxon among other basal members of the Teredinidae. The addition of Tamilokus raises the total number of genera in this economically and environmentally important group of wood-boring bivalves to 17.

Supplemental Information

Supplemental Information 1 Tamilokus mabinia FASTA 18S Sequence Data

Raw 18S sequence data. Please note, these files are for the purpose of review only. GenBank accession numbers will be provided upon acceptance of manuscript.

Click here for additional data file.

Supplemental Information 2 Tamilokus mabinia FASTA 28S Sequence Data

Raw 28S sequence data. Please note, these files are for the purpose of review only. GenBank accession numbers will be provided upon acceptance of manuscript.

Click here for additional data file.

Figure S1 Phylogenetic position of Tamilokus mabinia among the Teredinidae:

Bayesian analysis of the concatenated 18S and 28S nuclear rRNA gene sequences obtained from specimen PMS-3943P. The tree was constructed using the taxa presented in Distel et al. (2011). Numbers at nodes indicate posterior probabilities. Scale bars denote nucleotide substitutions per site.

Click here for additional data file.

The authors would like to thank Adam Baldinger and Jennifer Lenihan-Trimble (Museum of Comparative Zoology at Harvard University for use of the Keyence Microscope and Bruker Micro-CT. The work was completed under supervision of the Department of Agriculture-Bureau of Fisheries and Aquatic Resources, Philippines (DA-BFAR) in compliance with all required legal instruments and regulatory issuances covering the conduct of the research.

Additional Information and Declarations

Competing Interests

Author Contributions

Field Study Permissions

DNA Deposition

Data Availability

New Species Registration

Gary Rosenberg is an Academic Editor for PeerJ.

J. Reuben Shipway and Marvin A. Altamia conceived and designed the experiments, performed the experiments, analyzed the data, contributed reagents/materials/analysis tools, prepared figures and/or tables, authored or reviewed drafts of the paper, approved the final draft.

Gary Rosenberg, Gisela P. Concepcion and Margo G. Haygood conceived and designed the experiments, analyzed the data, contributed reagents/materials/analysis tools, authored or reviewed drafts of the paper, approved the final draft.

Daniel L. Distel conceived and designed the experiments, analyzed the data, contributed reagents/materials/analysis tools, prepared figures and/or tables, authored or reviewed drafts of the paper, approved the final draft.

The following information was supplied relating to field study approvals (i.e., approving body and any reference numbers):

The work was completed under supervision of the Department of Agriculture-Bureau of Fisheries and Aquatic Resources, Philippines (DA-BFAR) in compliance with all required legal instruments and regulatory issuances covering the conduct of the research. All the specimens used in this study were collected using DA-BFAR Gratuitous Permit GP-0140-17.

The following information was supplied regarding the deposition of DNA sequences:

GenBank: https://www.ncbi.nlm.nih.gov/nuccore/MH974682,

https://www.ncbi.nlm.nih.gov/nuccore/MH974683.

The following information was supplied regarding data availability:

Shipway, Reuben (2018): Tamilokus mabinia MicroCT Scan.avi. figshare. Media. https://doi.org/10.6084/m9.figshare.6967454.v1.

The following information was supplied regarding the registration of a newly described species:

Publication LSID: urn:lsid:zoobank.org:pub:0235E48D-D7B7-4A4B-A078-4204C1907240

Tamilokus mabinia LSID: urn:lsid:zoobank.org:act:88D193B6-9062-48C2-AC9D-EC6F858B3C05.

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
