# Peer review of "Tamilokus mabinia, a new, anatomically divergent genus and species of wood-boring bivalve from the Philippines"

_PeerJ, doi:10.7717/peerj.6256_

## Round 0.1 · original submission · Major Revisions

Even though two reviewers suggested only minor changes, a third reviewer has suggested quite significant work is needed, which I think will significantly improve the manuscript. Please address all the issues raised by the reviewers and the few comments I have listed below.

Please indicate in the text if any genera (and how many) were not examined for length of style and style sac (and other key traits). Please also indicate which species are used for measurements in Table 2.

Phylogenetic analysis - please state explicitly what proportion of currently accepted genera/species are included in the tree.

Given the level of polyphyly of several existing genera and incomplete taxon sampling I would tend to agree with the third reviewer that it may be too soon to erect a new genus without having established the boundaries of existing genera. If you decide to keep the new genus, then please add a sentence mentioning the problem of polyphyletic genera.

L170: 'without specifying any priors' - this is not correct. Your choice of model etc are all priors - you have, however, quite correctly, left them free to vary and not set any parameter values.

Table 1: I agree with Reviewer 3 that as it is, Table 1 does not serve much purpose. I suggest you delete the locality, co-ordinate and depth columns and include these details in the figure legend and instead give final voucher numbers for all types and link each one explicitly with any associated morphological and molecular data giving figure references. Please also check co-ordinates - it looks as thought cells have been copied in an excel file as each one goes up by 0.001 degrees.

Please italicise names in tree and indicate specimen number for new sample.

Please add a note to the figure legend for Fig. 6 and Supplementary data 1 to explain the comment ‘possibly L. floridanus’

·

Basic reporting

No comment.

Experimental design

No comment.

Validity of the findings

No comment.

Additional comments

In this study the authors combine anatomical, morphological and molecular data to provide compelling evidence of the existence of a new genus and species in the Teredinidae. The ms is in general clearly written. Below are some minor comments and suggestions on some specific points of your ms. I hope you find them helpful.
Line 82- There are no accession numbers in Table 1.
Line 169-170 “under accession numbers reported in Table 1”. Again, there are no accession numbers in Table 1. A column for them should be already there although they might be supplied at a later stage.
Line 214- “Kidney bean shaped stomach” is there a semicolon missing before stomach?
Line 224- After “faeces” shouldn’t it be a semicolon instead of a comma?
Line 224- Does Tamilokus have more than one caecum? Check if “ceca” is correct here.
Line 228- misspelling for “caecum”. There is also a space too much before the semicolon.
Line 300- Should be “(…) rRNA gene sequences”. As it is, it implies that RNA was sequenced, which I think it is not the case.
Line 238- Different spelling for caecum than in line 228. Should be uniform throughout the text.
347-348- Instead of “intestines” should be "intestine" as in line 373.
Line 368-372 is repetition of line 300-304. It should be rephrased in the Discussion to avoid repetition.
Line 401- Take “and” before “intestine”.
Line 401- Again check the use of “ceca”.
Line 407-411- In my opinion this new information should be in the Introduction and not in the conclusions. In the conclusions the authors shouldn’t bring in anything new.
Line 413- Treneman et al. 2018 do not describe a new species. This reference can be added to line 414 where it refers that additional works suggest new species.

Almost all references I looked at have issues with the format. This section needs attention.

Figure 2- commas after abbreviations not uniform. In addition, “ca” and “gi” are missing from the list.
Figure 5- After “A” there is a comma and after “B” a dash. It needs to be uniform.
Table 1- There is no reference to accession numbers.

·

Basic reporting

This is a well developed and clearly structured manuscript.

Experimental design

The range of morphological and molecular techniques is appropriate and the use of CT techniques particularly interesting in this context.

Validity of the findings

The conclusions are well stated and supported.

I noticed a few technical issues that I suggest addressing:

The most important one is the need for better documentation of the voucher and type material. Lines 105-108 state that voucher specimens are at ANSP and the Ocean Genome Legacy Center. The type material is described (lines 269-272) as having “PMS” numbers, with the holotype to be deposited in Manila and paratypes “deposited at ANSP, MCZ and OGL.” The type locality is stated to have coordinates “13.758, 120.925” (line 274). This refers, minimally, to the collecting site of the holotype, which is explained (line 269) to be specimen number “PMS-3916Y.” Table 1, meanwhile lists eight specimens (7 in Manila, one at OGL -- none at ANSP or MCZ) and, most importantly, indicates that the holotype specimen PMS-3916Y is not from the given type locality coordinates. Please provide a comprehensive listing that links the “PMS” numbers (and any other potential vouchers) referred to in this manuscript to the acronyms and museum collection numbers of the various repositories. Also, please identify the type specimens in the figure captions.

Collecting sites and Figure 1 (map): I cannot discern any individual collecting sites in the map. Table 1 gives a range of different, but adjacent, coordinates for the specimens (was this a transect?). Can you plot them on the map?

I suggest including some additional explanation with the new genus and species names: I assume these are of “masculine gender” and “used as a noun in apposition,” respectively?

Anatomical measurements: On how many specimens where the statements based? Please add “n=” where appropriate.

Table 3: The author/year information of genus-group taxa should not be in brackets or parentheses.

The paper is very well written – I only encountered one part that I could not quite interpret (“unique calcareous pallets which cannot be characterized within any existing genera” – lines 29-30). Is this just meant to emphasize that they are unique within the family, or is there something particularly difficult in the character assessment for the other genera?

Reviewer 3 ·

Basic reporting

This manuscript purports to describe a new taxon of teredinid from the Philippines, complete with soft parts and CT Scans. Although it shows the internal anatomy and provides genetic contrast with other named members of the family, additional work is essential. The text needs to be revised to eliminate the grammatical errors (e.g. line 58) and mis-statements such as that the shell is covered by ridges (line 61). Phrases such as “the anterior excavation face” (line 64) should be simplified. The references are not cited either alphabetically or chronologically in the text, in the few instances I checked, the year cited in the text are not the same as those provided in the references. Papers that are less than appropriate are cited, for instance Distel et al. 2011 is not a great reference for the economic impact of the teredinids. The format of the references differs within the section. PMS, apparently the specimen depository, is not defined. Not all the abbreviations shown on the figures are defined in the captions. The map figure doesn’t identify the island as being in the Philippines, it is cropped too far. Is the scale bar in each figure the same for each of the images in that figure? “Variation” is used where I suspect they mean “difference”. Line 103 “were utilized for anatomical characterization” could you please say this more simply?

Line 105 you say where voucher specimens deposited, but not where the type specimens are deposited. You say you used partial specimens for anatomical studies & DNA making the identification of those partial specimens based on the pallets. So you never actually looked at what you cite as key characters in the type specimens?

The Diagnosis isn’t a diagnosis. A diagnosis should succinctly state the characters that can be used to identify a specimen of the taxon without reference to anything else. Usually it is written in telegraphic style. I don’t know what a “Definition” section is in reference to the taxonomic literature. There should be a “Remarks” section in which one contrasts the new taxon to those overtly similar or with which the new taxon could be confused. Where is the new species actually named? Line 197 “mid-point of distal shell valves” is unclear. As is “ending hook-shaped posterior in the stomach” Line 204

Teredora & Uperotus are suggested to be open ocean animals – does this impact your understand and the comparison?

Those issues must be fixed before the manuscript can be considered for acceptance, but two additional issues must be raised. The authors here make the case that this taxon is strongly different from those heretofore named, but it appears that it differs in characters that have not been cited in taxonomic treatments, e.g. the crystalline style length, the soft parts associated with the valve. This creates a problem, as who knows what the character states are among taxa with older taxonomic names characters?
In terms of the style length, the authors present one set of measurements for each of 9 taxa, but report the source only once in the methods. Turner 1966 whether she fully documented the style is unknown. It would be better to measure multiple individuals and calculate allometric relationships. Ratios are extremely suspect and should be avoided. In re. Turner 1966, Fig. 5C labels the Mantle Collar on T. malleolus although the present manuscript says that taxon does not have a mantle collar.

The authors appear to weigh the shape of the cecum more strongly than any other character (line 323 which needs a citation!) apparently because this character state is unusual or at least not often reported. Table 3 reports diagnostic features of taxa sharing this character state. The abstract however begins with mention of the Cephalic hood. Pallet shape has been the character of choice in the family. How does the new taxon compare to taxa which share overtly similar pallets? Remember please pallet morphology can change over time… My point here is that the previous names were erected on a set of characters; you are devising new characters that justify erecting a taxon without addressing the traditional taxa. No one would say that it isn’t high time something was added to the character matrix, but rejecting or worse yet, ignoring, the older characters without considering them is inappropriate. List comparative material examined and their character states.
The phylogeny shown in the figure indicates that this taxon is different from all others in the matrix, but given paraphyletic genera in the tree, this information is less than compelling. It makes me wonder whether creating another monotypic genus is going to contribute to resolving the group’s phylogeny. There is somewhere in a Turner paper a depiction of coloration and complex cilia in siphonal openings.

Table 1 is hardly worth having, this information can be put into the text, or indicated on a modified Fig. 1.

The paragraph from Lines 350-362 attempts to argue polarity of character states from the phylogeny in Fig. 6 rather than from outgroup comparison. This is inappropriate. The paragraph from lines 364-375 hardly contributes anything to the manuscript. Why is it here?

The authors need to look at this manuscript carefully. Double check all the statements, correct the references or find ones that are more appropriate. Rethink the approach – and the format of the manuscript. Perhaps read a few species descriptions in peer-reviewed journals to better understand how to go about preparing one. If exclusive reliance on literature sources is absolutely necessary, then make sure that you look at those references carefully and thoroughly. Describing species is not easy, and when you describe a species from the Philippines please be aware that most readers will not be native English-speakers, requiring that you be as clear as possible.

Experimental design

none.

Validity of the findings

This probably is a new species. As to new genus, that level is arbitrary. Is it so well described that I could identify a specimen from the wild? no.

Additional comments

Pay attention to the details to make the manuscript as good as it can be.

---

## Round 0.2 · Minor Revisions

Reading through the reviewer's comments I would ask that you please describe the pallets in the description and follow the formal taxonomic outline and pattern of reporting suggested by the reviewer. This should only require minor edits so hopefully it wont take too long. If you can get the manuscript back to me with all these changes before the 18th December I will deal with it before Christmas.

Reviewer 3 ·

Basic reporting

text could benefit from cutting out redundancies.

Experimental design

No comment

Validity of the findings

it's apparently new. Is it a contribution to the field and the best it can be is the question.

Additional comments

This manuscript still needs revision. The point of the description is to allow easy differentiation of this taxon from others. The characters that have been used for sp. identification in this family relate to the pallets. The description of the genus does not mention the pallets. It therefore defines the group on characters that are unknown in other taxa. This unnecessarily complicates its utility for future workers.
The authors should imagine themselves as a non-English speaking marine biologist who has found a teredinid. Painstakingly, the biologist translates every word of the description and compares it to the unknown. They may have prepared the pallets to allow comparison to previously named taxa, but, faced with this description are stuck with having to prepare the guts, without a micro-CT Scanner available… without knowing the anatomy. What characters can they look at quickly & simply to put this in this taxon? That information is the diagnosis. See this work Lyrodus mersinensis sp. nov. (Bivalvia: Teredinidae) another cryptic species in the Lyrodus pedicellatus (Quatrefages, 1849) complex. A full treatment of multiple characters goes into the description. A discussion of how this taxon compares to other taxa that are most similar to the new one goes in the Remarks. In addition when comparing the new taxon to others, one should list the material examined, that is what specimens did you use for the direct comparison?
As a generality in the description proper, because we are conchologists, the discussion of hard parts typically goes first, then any discussion of soft parts.
Authorities should be cited with a comma.
If your statements about other genera are all based on Turner’s 1966 monograph – which treated lots more than taxonomy - cite that work. If they are based on your examination of specimens, cite the specimens examined.
The authors need to follow journal instructions about citing the Zoobank information: Include the name.
Does MCZ have a “Turner Collection”?
The description of the genus and species are about 110 lines including double hard returns, and the discussion is another 100 lines. It should be shorter, as it begins by saying the new taxon is distinct from all others. Well, yes – you feel this otherwise you would not have attempted the description. In addition, there is no need to repeat characters in the discussion, it is not a re-hash of the description. Just emphasize what to you are the interesting features that might be of interest to someone else. Don’t repeat the self-evident.
Why not list the type specimens with their museum numbers in association with the description? One assumes even if they are catalogued at ANSP those numbers will always reflect their present status that is post transfer to their native country.
In the text there is consistent mention that this taxon differs from previously named taxa. This is risky as many of those names are now synonymized and their soft parts are unknown. You can say that this is distinct from all currently accepted taxa, but there are more synonymized taxa (many of which do not have soft parts known) than there are accepted ones.
Lines 299-306 go from defining the cephalic collar, to saying its use is unknown, to proposing a use for it. This could easily be shortened.
Line 326 needs a reference.
A manuscript with an abstract at this length does not also need conclusions.
You report the apparent association of large labial palps with filter-feeding, but regardless of their ability to do so, they also function one would think to move wood shavings into the gut. Small or large, they need to aid ingestion of whatever.
How was pallet in Fig. 4 prepared? Dry? Alcohol stored?
Report doi for references
Your expressed expectation that a reviewer provide you with a reference is demeaning.
The bottom line is that this is likely a new taxon. Is this description as clear and concise as it can be? Is it a contribution to the field? Just because you have micro-CT data that no one else has had, doesn’t mean that it helps with all those previous species descriptions that lack such data.

---

## Round 0.3 · accepted · Accept

Thank you for making the last few changes to your manuscript. I hope you and all your co-authors have a very happy Christmas!

#